# The Association between the Histological Subtypes of Mesothelioma and Asbestos Exposure Characteristics

**DOI:** 10.3390/ijerph192114520

**Published:** 2022-11-05

**Authors:** Trudie Vorster, Julian Mthombeni, Jim teWaterNaude, James Ian Phillips

**Affiliations:** 1Faculty of Health Sciences, Department of Biomedical Sciences, University of Johannesburg, Johannesburg 2028, South Africa; 2Pathology Division, National Institute for Occupational Health, National Health Laboratory Service, Johannesburg 2000, South Africa; 3Diagnostic Medicine, Cape Town 7708, South Africa

**Keywords:** mesothelioma, subtypes, PATHAUT, asbestos type, asbestos burden

## Abstract

Asbestos mining operations have left South Africa with a legacy of asbestos contamination and asbestos-related diseases continue to be a problem. The large-scale mining of three types of asbestos presents a unique opportunity to study malignant mesothelioma of the pleura (mesothelioma) in South Africa. This study aimed to describe the demographics of deceased individuals diagnosed with mesothelioma and explore any associations between the histological morphology of mesothelioma and asbestos characteristics. We reviewed the records of all deceased miners and ex-miners from the Pathology Automation System (PATHAUT) database of the National Institute of Occupational Health (NIOH) that were histologically diagnosed with mesothelioma in the period from January 2006–December 2016 (11 years). The study population does not include all cases of mesothelioma in South Africa but rather those that reached the compensation system. Crocidolite asbestos fibres were identified in the majority of mesothelioma cases (*n* = 140; 53.4%). The epithelioid subtype was most commonly present in both occupational and environmental cases. Cases with the sarcomatous subtype were older at death and fewer female cases were diagnosed with this subtype. No relationship between mesothelioma subtype and asbestos type or asbestos burden or fibre size was established.

## 1. Introduction

South Africa is a uniquely mineral-rich country, and its economy is underpinned by mining. Asbestos was mined for more than 100 years [1]. South Africa was the world’s third largest producer of asbestos, with production peaking in 1977 [2]. Uniquely, three different types of asbestos, namely amosite, crocidolite, and chrysotile were mined on a large commercial scale [3]. Most of South Africa’s asbestos production was exported but some was used to manufacture goods within the country [4].

Asbestos has useful properties and confers strength and durability to the manufactured products in which it is incorporated [3]. However, asbestos is associated with adverse health effects. Inhalation of asbestos fibres can cause diffuse pleural thickening, pleural plaques, asbestosis, lung cancer, and malignant mesothelioma of the pleura (mesothelioma) [5]. The association between crocidolite asbestos and mesothelioma was established by Wagner in 1960 while working at what is now the NIOH [6]. Mesothelioma is an invariably fatal tumour which usually occurs about 20 to 40 years after exposure to asbestos [7]. The diagnosis is histologically confirmed and classified under three main histological subtypes of mesotheliomas, namely the epithelioid subtype, sarcomatous subtype, and mixed or biphasic subtype with both epithelioid and sarcomatous elements [8]. Current treatment modalities are ineffective [9].

Due to the adverse health effects associated with asbestos exposure, countries began to ban its import and use. The demand for asbestos decreased and South African mines began to close [10]. The last asbestos mine in South Africa closed in 2002 [10]. Six years later South Africa joined some 67 countries in banning asbestos [11].

Because of activities, such as mining, milling, and transporting asbestos, there was widespread contamination of the environment, which persists in some areas. In addition, there are large amounts of asbestos in manufactured items in the built environment, such as asbestos cement roofs [12]. This legacy of asbestos in the South African environment means that there is a potential for exposure and disease for many years to come. South Africa is amongst 10 countries with the highest burden of mesothelioma in the world [1].

Exposure to asbestos can be grouped into occupational and non-occupational exposure. Occupational exposure includes asbestos miners, millers, and workers involved in the manufacturing of asbestos products. Non-occupational exposure can be grouped into domestic, neighbourhood, and true environmental exposure. Domestic exposure can also be referred to as para-occupational or familial exposure. This occurs when asbestos workers carry asbestos fibres home on their working clothes, which usually affects their family members. Neighbourhood exposure, also referred to as environmental exposure, affects residents living close to mine tailings or other asbestos-contaminated areas. True environmental exposure arises from naturally occurring asbestos-contaminated soil [13].

Mine workers in general may be compensated for asbestos-related diseases in terms of the Occupational Diseases in Mines and Works Act 78 of 1973 (ODMWA, 1973) [14]. In addition, mesothelioma amongst asbestos mine workers or those environmentally exposed to asbestos mines may be compensable through the Asbestos Relief Trust (ART) or the Kgalagadi Asbestos Relief Trust (KRT) [15]. These compensation systems facilitate the submission of the lungs of deceased patients to the NIOH for examination and diagnoses to assist with the compensation process. Data concerned with the examination of the lungs are recorded and stored at the NIOH.

The large-scale mining of three types of asbestos along with the detailed data captured at the NIOH, presents a unique opportunity to study mesothelioma in South Africa. The aim of this study was to describe the data collected on individuals diagnosed with mesothelioma at the NIOH. The associations between the asbestos type, fibre size, and burden and mesothelioma subtype were also analysed.

## 2. Materials and Methods

### 2.1. Study Population and Data collection

This was a cross-sectional study of all deceased miners and ex-miners histologically diagnosed with mesothelioma at autopsy, whose data had been captured on the PATHAUT database for the period from January 2006–December 2016 (11 years). Data of individuals diagnosed with mesothelioma include demographics, such as age, sex, employment histories, and pathology findings of the respiratory organs examined.

Data concerning the asbestos lung fibre burden, asbestos fibre counts, and asbestos type, fibre size, and concentration were obtained from the Electron Microscopy (EM) Unit database within the NIOH Pathology Department. The lung fibre burden is determined by extracting the asbestos fibres from the lungs. The asbestos fibre type, fibre size, and asbestos body concentration are determined by SEM together with energy dispersive spectroscopy (EDS). This method was previously described by Phillips and Murray 2010 [16].

PATHAUT was the main source of data and the EM records were linked to the main source to obtain additional information on compensation outcomes and asbestos lung fibre burden.

### 2.2. Variable Descriptions and Data analysis

Statistical analysis was performed using Stata version 16 (StataCorp, College Station, TX, USA) statistical software. Descriptive statistics and tabulation were conducted for the demographic characteristics, such as age, sex, region, employment history, and commodity exposure, as well as the compensation outcomes. Bivariate analysis was conducted to determine the relationship between demographic and regional variables and the asbestos exposure types (occupational or environmental). Analysis involving categorical variables, such as sex, region, and exposure types, was conducted using Pearson’s chi-squared test (Fisher’s exact test was utilised when the expected frequency was less than 5 in more than 25% of the cells). The mean difference in continuous variables (such as age) across exposure types (occupational or environmental) was assessed using Student’s independent *t*-test. Similarly, the chi-squared test and Student’s *t*-test were utilised to assess the relationship between categorical and continuous variables, respectively, and the compensation outcome.

The asbestos lung fibre burden was described in terms of the asbestos type and fibres per gram of dry weight of lung tissue. The asbestos fibre sizes were described in the ranges of 1–5 µm, ≥5–10 µm, and >10 µm. Asbestos burden or concentration of fibres per gram of dry weight of lung tissue was described in ranges (1–999,999, 1,000,000–2,999,999, and ≥3,000,000). The mean and standard deviation, range, frequency, and percentages were utilised to describe the type, fibre size, and burden of asbestos in the lungs. The relationship between the aforementioned explanatory variables and asbestos exposure (occupational and environmental) was conducted using Student’s *t*-test and the chi-squared test (or Fisher’s exact test in appropriate conditions) for the continuous and categorical variables, respectively. Equality of variance was also assessed before conducting either equal Student’s *t*-test or unequal Student’s *t*-test as appropriate.

The histological morphology of mesothelioma was described for the epithelioid, sarcomatous, or biphasic (mixed) subtypes. The histological subtype was the dependent variable and the explanatory variables were the asbestos type, fibre size, and the asbestos fibre burden. The relationship between mesothelioma subtypes (categorical) and asbestos type (categorical), fibre size (categorical), and asbestos burden ranges (categorical) was assessed using Pearson’s chi-squared tests. Furthermore, a one-way analysis of variance (ANOVA) was conducted to assess the differences in the mean levels of continuous variables, such as asbestos fibres, across the three categories of histological subtypes of mesotheliomas. A post hoc Bonferroni test was conducted when the *p*-value of ANOVA was statistically significant to determine where the pairwise difference(s) lie. A two-tailed test of the hypothesis was assumed and a *p*-value < 0.05 was assumed to be statistically significant.

Ethical approval was obtained from the Research Ethics Committee at the University of Johannesburg prior to the commencement of the study.

## 3. Results

### 3.1. Demographic Data

In the 11-year period, 270 cases of mesothelioma were identified in the PATHAUT database. Of the 270 mesothelioma cases, 89.3% were occupationally exposed to asbestos and 10.7% were environmentally exposed. The mean age of all of the individual mesothelioma cases was 64.0± 10.8 years. Furthermore, the mean age of the occupationally exposed cases was similar to the mean age of the environmentally exposed cases (occupation vs. environment: 64.0 ± 10.8 years vs. 64.3 ± 10.8 years, *p* = 0.88). Nearly four-fifths of the mesothelioma cases were males (*n* = 214, 79.3%). The majority of the occupationally exposed cases were male patients (86.3%, *n* = 208), while the majority of the environmentally exposed cases were female (65.5%, *n* = 19) Table 1. There was no statistically significant difference between the mean age among male and female mesothelioma patients (male vs. female: 63.8 ± 10.9 years vs. 64.9 ± 10.3 years, respectively, *p* = 0.52).

The mean years of exposure among the environmentally exposed cases (23.5 ± 2.1 years) were higher than the mean years of exposure among the occupationally exposed cases (13.4 ± 11.3 years), as shown in Table 1.

### 3.2. Exposure Data

Occupational cases mostly worked in different mineral mines, which resulted in mixed exposures. However, the exposure data are based on the longest service history. In nearly half of the occupational cases (46.5%, *n* = 112), their longest service was in the asbestos mining industry with a mean length of service of 5.8 ± 6.5 years, as presented in Table 2. The mean length of exposure in Table 2 for all other commodities was measured in double digits.

### 3.3. Asbestos Fibre Burden

Of the 270 mesothelioma cases identified, 97.0% (*n* = 262) were sent for asbestos fibre analyses. Of these, 90.8% (*n* = 238) were occupational cases and 9.2% (*n* = 24) were environmental cases. Crocidolite alone was found in the lungs of about half (*n* = 140, 53.4%) of the mesothelioma cases and no asbestos fibre was identified in about one-third of cases (*n* = 98, 37.4%). There was no statistically significant difference in the pattern of asbestos fibres that were found among the environmentally and occupationally exposed cases (*p* = 0.42), as shown in Table 3. Of the cases that contained only crocidolite fibres, 92.1% (*n* = 129) were occupational and 7.9% (*n* = 11) were environmental cases.

### 3.4. Histological Features of Mesothelioma

The histological features of the mesothelioma cases are presented in Table 4. Nearly two-thirds (*n* = 64.4%, *n* = 174/270) of the mesothelioma cases were of the epithelioid histological subtype. The next common histological subtype was the biphasic subtype (23.3%, *n* = 63/270) (Table 4) Furthermore, the predominant histological subtype among the environmentally exposed (69.0%) and occupationally (63.9%) exposed patients was the epithelioid subtype. There was no statistically significant difference in the histological pattern among the environmentally exposed cases as compared to the occupationally exposed cases (*p* = 0.65), as presented in Table 4.

The mean age of individuals with different histological subtypes of mesotheliomas is displayed in Table 5. Individuals with the sarcomatous subtype had the highest mean age of 68.8 ± 9.4 years while individuals with the biphasic subtype had the lowest mean age of 62.8 ± 9.9 years. The one-way analysis of variance showed that there was a statistically significant difference in mean age across the histological subtypes, *p* = 0.02. The post hoc Bonferroni test showed that the difference was between the mean age of individuals who had sarcomatous and biphasic subtypes (68.8 ± 9.4 vs. 62.8 ± 9.9, respectively, *p* = 0.03) and between individuals with the sarcomatous and epithelioid histological subtypes (68.8 ± 9.4 vs. 63.6 ± 11.2, respectively, *p* = 0.03). There was no statistically significant difference in the mean age of individuals who had biphasic and epithelioid histological subtypes (62.8 ± 9.9 vs. 63.6 ± 11.2, respectively, *p* = 1.00). Furthermore, there was no statistically significant difference in the pattern of the histological subtypes of mesothelioma among females as compared to males (*p* = 0.65), as shown in Table 5.

### 3.5. Fibre Burden per Mesothelioma Subtype

The mean concentration and range of fibres by exposure type are presented in Table 6. The mean concentration of asbestos fibres in the lungs of mesothelioma patients was 2,939,321 ± 9964 million per gram of dried lungs. Generally, the mean concentration of the various types of asbestos bodies and fibres was higher among the lungs of mesothelioma patients that were designated as occupationally exposed as compared to the lungs of patients with environmental exposure (Table 6).

Table 7 shows the pattern of asbestos bodies and fibre counts by histological subtype among the environmentally and occupationally exposed patients. Of the occupational cases, the epithelioid subtype had the highest mean number of asbestos fibres, asbestos bodies, and crocidolite fibres. The highest mean number of amosite fibres occurred among the biphasic subtype. However, the one-way analysis of variance showed that the mean concentration of the various asbestos fibres was not statistically different across the histological subtypes. Of the environmental cases, the epithelioid subtype had the highest average number of asbestos fibres and the highest average number of crocidolite fibres. Asbestos bodies were only identified in the epithelioid subtype.

The mean concentration of fibres categorized by size is displayed in Table 8. The analysis of variance showed that there were no statistically significant differences in the sizes of the asbestos fibres across histological subtypes.

### 3.6. Relationship between Asbestos Type, Fibre Size, Burden, and Mesothelioma Subtype

Of the 270 mesothelioma cases in this study, 164 cases were used to establish the relationship between the mesothelioma subtype and asbestos type. These cases had identified both a mesothelioma subtype and asbestos type. The relationship between asbestos fibres and the histological subtype of mesothelioma is displayed in Table 9. There was no statistically significant relationship between the asbestos fibres and the histological subtypes (*p* = 0.51). For the majority of the mesothelioma cases, the lungs had similarly high proportions of crocidolite across the histological subtypes (epithelioid vs. biphasic vs. sarcomatous: 85.3% vs. 88.9% vs. 78.9%, respectively, *p* = 0.51).

The relationship between asbestos fibre size, the number of fibres, and mesothelioma subtype are presented in Table 10. For the crocidolite fibre size of 1–5 µm and asbestos fibre amounts of 0, 1–999,999, 1,000,000–2,999,999, and ≥3,000,000, there was no statistically significant association across the histological subtypes (*p* =0.64). The same was observed for crocidolite > 5–10 µm (*p* =0.46) and crocidolite > 10 µm (*p* = 0.84).

For the amosite fibre size of 1–5 µm and asbestos fibre amounts of 0, 1–999,999, 1,000,000–2,999,999, and ≥3 000 000, no significant association was found between the histological subtypes (*p* = 0.68). Similarly, there was no statistically significant relationship between amosite > 5–10 µm (*p* = 0.75) and amosite > 10 µm (*p* = 0.34) and the histological subtypes.

## 4. Discussion

The aim of this study was to describe the data collected on individuals diagnosed with mesothelioma at the NIOH from 2006 to 2016. This was achieved by describing and comparing the environmental and occupational cases of mesothelioma in terms of demographic characteristics and the asbestos fibre burden. The associations between mesothelioma subtype and asbestos type, burden, and fibre size were also analysed.

The mesothelioma cases extracted from the PATHAUT database over the 11-year period only present cases that came through the NIOH for diagnosis as part of the compensation process. A study reported trends of mesothelioma cases among South Africans during a similar period and cases reported in PATHAUT equate to about 18% of all reported mesothelioma cases in South Africa [17]. There are several barriers to requesting and providing consent for autopsy by relatives of deceased miners. This was previously described in 2017 [18]. Of the cases diagnosed with mesothelioma in the PATHAUT database over the 11-year study period, there were more occupational than environmental mesothelioma cases. In the occupational setting, more than 80% of the cases were male. Women comprised up to half of the South African asbestos mine workforce; they were invisible in the industry as women were never formally employed in asbestos mines and hence not registered [19]. This could explain why so few women were identified in this study, especially in the occupational setting. Some may have been included as environmental cases.

We found that the mean age of individuals who had mesothelioma was about 64 years and the peak prevalence of mesothelioma among our study population was 50–59 years. Our result was in agreement with the findings of a global study that reported a mean age of 63.4 years among mesothelioma patients in South Africa [20]. However, the study reported a higher global mean age of mesothelioma deaths of 70.1 years and ascribed the lower mean age of mesothelioma cases in South Africa to a background lower life expectancy as compared to the life expectancy indices in high-income countries [20]. The lower mean age of South Africa’s mesothelioma cases can be attributed to competing causes of death, with fewer people reaching older ages, having succumbed to other causes [21].

Although mesothelioma is described as a disease of the elderly [22], younger patients also develop the disease. Our study found that the youngest mesothelioma case was 40 years old. Mesothelioma has been reported in the literature, such as that of a 17-year-old boy in Mexico [23]. Mesothelioma among young adults in South Africa may be partly related to the common practice of early exposure when children play on asbestos waste dumps or used these dumps as sandpits [24].

Asbestos fibre analysis showed that crocidolite was present in the lungs of more than half (53.4%) of the cases. Only one case contained chrysotile fibres and these were found together with amosite fibres. No cases were reported where only chrysotile fibres were present in the lungs. These findings are in keeping with a study that found most cases contained crocidolite and chrysotile was never found alone [25]. Previous studies showed that chrysotile is cleared from the lungs by macrophages, even though it may still cause damage to the lungs [26,27,28]. A more recent study also showed that chrysotile is cleared from the lungs, which would explain why so few chrysotile fibres were found when performing the fibre analysis [29,30]. No asbestos fibres were identified for 37.4% of cases. Some of these cases were described as the lungs completely being replaced by tumour where in others only asbestos bodies were identified.

The epithelioid subtype of mesothelioma was the predominant histological subtype with a prevalence of 64.4% among the mesothelioma cases. Furthermore, the epithelioid subtype was also a major histological subtype among both occupational (63.9%) and environmental (69.0%) cases. This histological pattern that was observed among our study population is similar to other studies’ findings that reported the epithelioid histological subtype to be the major subtype of mesothelioma [8,31,32]. Only three female cases (9.1%) were diagnosed with the sarcomatous subtype compared to 20.6% and 19.0% for the biphasic and epithelioid subtypes, respectively.

Our study demonstrated that, on average, individuals with the sarcomatous subtype appeared to be about five years older at diagnosis than individuals diagnosed with the other two histological subtypes. Similarly, another study showed that patients diagnosed with sarcomatous mesothelioma were slightly older at diagnosis, thereby suggesting a longer latency period [8]. Another study also found a significant difference between the survival periods when comparing the epithelioid and sarcomatous subtypes. The mean survival period for the epithelioid subtype was 12.2 months versus 7.3 months for the sarcomatous subtype [31].

Expectedly, we found that the concentration of asbestos fibres was generally higher among the occupational cases of mesothelioma as compared to the environmental cases. The most commonly identified fibre in the lung (occupational and environmental cases) was crocidolite. However, there was no statistically significant difference in the asbestos fibres across the mesothelioma subtype. This finding is similar to another study that reported no strong evidence between asbestos exposure indicators and the histological subtype of mesothelioma [8,31]. In contrast, another study observed that the lungs of sarcomatous cases contained higher amosite concentrations [33].

## 5. Conclusions

No strong evidence was found to support any relationship between mesothelioma subtype and asbestos type, fibre size, or asbestos burden. However, there was a statistically significant difference in mean age between the sarcomatous and biphasic subtypes and between the sarcomatous and epithelioid subtypes.

Mesothelioma subtypes have different characteristics, prognoses, and outcomes. Research to further characterize the aetiopathology of these histological subtypes is necessary.

## Figures and Tables

**Table 1 ijerph-19-14520-t001:** Demographic characteristics, region, and annual mesothelioma diagnosis stratified by asbestos exposure.

Characteristics	Asbestos Exposure		Total N = 270 (%)	*p*-Value
	Environmental, *n* = 29 (%)	Occupational, *n* = 241 (%)		
Age (Mean ± SD) years	64.3 ± 10.8	64.0 (± 10.8)	64.0 ± 10.8	0.88 ^$^
40–49	2 (8.3)	15 (6.3)	17 (6.5)	0.75 ^Ω^
50–59	5 (20.8)	81 (34.2)	86 (33.0)	
60–69	8 (33.4)	66 (27.9)	74 (28.4)	
70–79	7 (29.2)	54 (22.8)	61 (23.4)	
80 and above	2 (8.3)	21 (8.9)	23 (8.8)	
Sex				
Female	19 (65.5)	30 (12.5)	49 (18.2)	< 0.00 ^*
Male	6 (20.7)	208 (86.3)	214 (79.3)	
Unknown	4 (13.8)	3 (1.2)	7 (2.6)	
Length of exposure (Mean ± SD) years	23.5 ± 2.1	13.4 ± 11.3	13.5 ± 11.3	< 0.00 ^$^

* Statistically significant at *p*-value < 0.05. SD: Standard deviation. ^$^ Student’s *t*-test, ^ Fisher’s exact test, ^Ω^ Pearson’s chi-squared test.

**Table 2 ijerph-19-14520-t002:** Number of occupational cases by commodity most exposed to and mean exposure years.

Commodity Most Exposed to	Total Number of Cases (%)	Length of Exposure (Mean ± SD), Years
Asbestos	112 (46.5)	5.8 ± 6.5
Gold	25 (10.4)	20.6 ± 11.5
Platinum	25 (10.4)	17.9 ± 9.9
Manganese	21 (8.7)	17.9 ± 10.7
Industry	10 (4.1)	20.6 ± 14.1
Iscor	8 (3.3)	17.4 ± 8.8
Coal	7 (2.9)	18.9 ± 13.0
Diamond	7 (2.9)	14.2 ± 15.9
Eskom	5 (2.1)	31.6 ± 6.5
Iron	3 (1.2)	16.7 ± 11.0
SA Railways	3 (1.2)	22.0 ± 12.5
Unknown	7 (2.9)	0.0
* Other	8 (3.3)	12.9 ± 12.3

* Includes copper, iron, vanadium, quarry, silica smelters, steel, lead, and lime.

**Table 3 ijerph-19-14520-t003:** Type of asbestos fibre among mesothelioma cases based on the pattern of exposure.

Asbestos Type	Asbestos Exposure		Total N = 262 (%)	*p*-Value
	Environmental *n* = 24 (%)	Occupational *n* = 238 (%)		
Crocidolite	11 (45.8)	129 (54.2)	140 (53.4)	0.42 ^
Amosite and Crocidolite	0 (0.0)	15 (6.3)	15 (5.7)	
Amosite	1 (4.2)	7 (2.9)	8 (3.1)	
Amosite and Chrysotile	0 (0.0)	1 (0.4)	1 (0.4)	
No Asbestos fibres identified	12 (50.1)	86 (36.1)	98 (37.4)	

^ Fisher’s exact test.

**Table 4 ijerph-19-14520-t004:** Histological features of mesothelioma cases by exposure type.

Histological Subtype	Exposure Type		Total N = 270 (%)	*p*-Value
	Environmental *n* = 29 (%)	Occupational *n* = 241 (%)		
Epithelioid	20 (69.0)	154 (63.9)	174 (64.4)	0.65 ^$^
Biphasic	7 (24.1)	56 (23.2)	63 (23.3)	
Sarcomatous	2 (6.9)	31 (12.9)	33 (12.2)	

^$^ Pearson’s chi-squared test.

**Table 5 ijerph-19-14520-t005:** Pattern of age and sex across the histological subtypes of mesotheliomas.

Characteristics	Histological Subtypes			*p*-Value
	Epithelioid	Biphasic	Sarcomatous	
Age (mean ± SD) Years	63.6 ± 11.2	62.8 ± 9.9	68.8 ± 9.4	0.02 ^¥^
Sex				
Female	33 (19.0)	13 (20.6)	3 (9.1)	0.65 ^
Male	136 (78.2)	49 (77.8)	29 (87.9)	
Unknown	5 (2.9)	1 (1.6)	1 (3.0)	

^¥^ One-way analysis of variance test; ^ Fisher’s exact test.

**Table 6 ijerph-19-14520-t006:** Asbestos bodies and fibre counts by exposure type (millions per gm of dried lung).

Asbestos Characteristics	Occupational Mean ± SD, (Range)	Environmental Mean ± SD, (Range)	Total Mean ± SD, (Range)	^£^*p*-Value
Asbestos bodies	339,731 ± 1,166,446 (0–12,600,000)	45,389 ± 94,848 (0–386,645)	312,768 ± 1115 (0–12,600,000)	0.00
Asbestos fibres	3,180,292 ± 1,040,000 (0–92,100,000)	549,698 ± 1,824,964 (0–8,791,489)	2,939,321 ± 9964 (0–92,100,000)	0.00
Crocidolite	1,122,159 ± 4,086,234 (0–47,000,000)	268,396 ± 942,343 (0–4,576,008)	1,043,952 ± 3,911,657 (0–47,000,000)	0.01
Amosite	60,502± 488,711 (0–6442,069)	423 ± 2075 (0–101,69)	54,998 ± 466,024 (0–6,442,069)	0.06
Chrysotile	2.1 ± 40 (0–620)	0.0	2.6 ± 40.2 -	-

^£^ Student’s *t*-test.

**Table 7 ijerph-19-14520-t007:** Asbestos bodies and fibre counts per mesothelioma subtype (millions per gm of dried lung).

Asbestos Fibres	Occupational				Environmental			
	Epithelioid (Mean ± SD)	Biphasic (Mean ± SD)	Sarcomatous (Mean ± SD)	*p*-Value	Epithelioid (Mean ± SD)	Biphasic (Mean ± SD)	Sarcomatous (Mean ± SD)	^¥^*p*-Value
Asbestos bodies	418,617 ± 1,369,761 (0–12,604,909)	277,991 ± 8,046,345 (0–4,288,916)	62,471 ± 1,47,019 (0–584,099)	0.2735	64,079 ± 107,979 (0–386,645)	0 (0–0)	0 (0–0)	-
Asbestos fibres	4,305,487 ± 12,732,351 (0–92,100,633)	1,248,760 ± 3,234,999 0–16,833,994	1,090,117 ± 1,911,437 (0–6,124,898)	0.0851	728,206 ± 2,158,220 (0–8,791,489)	92,718 ± 197,146 (0–491,665)	256,936 ± 0 (0–256,936)	0.77
Crocidolite	1,542,616± 5,029,989 (0–46,955,632)	395,855 ± 937,251 (0–5,523,175)	349,166 ± 707,095 (0–2,372,658)	0.1075	359,574 ± 1,115,324 (0–4,576,008)	40,033 ± 83,076 (0–207,495)	88,536 ± 0 (0–88,536)	0.78
Amosite	4492 ± 318,188 (0–326,637)	124,272 ± 868,509 (0–6,442,069)	23,731 ± 96,925 (0–508,216)	0.5329	598 ± 2466 (0–10,169)	0.0	0.0	-
Chrysotile	4 ± 50 (0–620)	0.0	0.0	-	0.0	0.0	0.0	-

^¥^ ANOVA test.

**Table 8 ijerph-19-14520-t008:** Mean concentration of fibre sizes per mesothelioma subtype (millions per gm of dried lung, μg).

Occupational Cases									
**Asbestos Type**		**Crocidolite ***			**Amosite ***			**Chrysotile ***	
**Fibre Size ** **(µm)**	**1–5** **Mean ± SD**	**>5–10** **Mean ± SD**	**>10** **Mean ± SD**	**1–5** **Mean ± SD**	**>5–10** **Mean ± SD**	**>10** **Mean ± SD**	**1–5** **Mean ± SD**	**>5–10** **Mean ± SD**	**>10** **Mean ± SD**
Epithelioid	443,972 ± 1,578,011	644,023 ± 2,236,789	454,606 ± 1,493,316	20,485 ± 161,395	16,384 ± 126,439	8058 ± 49,272	0.0	0.0	4.1
Biphasic	101,576 ± 359,481	142,661 ± 335,643	151,608 ± 342,879	41,450± 302,087	32,539± 226,611	50,283 ± 340,046	0.0	0.0	0.0
Sarcomatous	52,238 ±127,165	139,176 ± 299,715	157,740 ± 327,523	0.0	3051 ± 14,620	20,679 ± 93,057	0.0	0.0	0.0
^¥^*p*-value	0.11	0.12	0.19	0.62	0.65	0.29	-	-	-
**Environmental cases**		**Crocidolite ***			**Amosite ***				
**Fibre size (µm)**	**1–5** **Mean ± SD**	**>5–10** **Mean ± SD**	**>10** **Mean ± SD**	**1–5** **Mean ± SD**	**>5–10** **Mean ± SD**		**>10** **Mean ± SD**		
Epithelioid	27,416± 90,113	109,150 ± 306,630	223,006 ± 806,913	0.0	598		0.0		
Biphasic	3556 ± 8710	20,963 ± 37,672	15,513 ±37,998	0.0	0.0		0.0		
Sarcomatous	0.0	59,024 ± 0	29,512 ± 0	0.0	0.0		0.0		
^¥^*p*-value	0.79	0.79	0.81	-	-		-		

* Fibres per µg dry weight; ^¥^
*p*-value of ANOVA.

**Table 9 ijerph-19-14520-t009:** Relationship between histological mesothelioma cases by asbestos fibre type and mesothelioma subtype.

Asbestos Fibres	Epithelioid *n*, %	Biphasic *n*, %	Sarcomatous *n*, %	Total *n*, %	*p*-Value
Crocidolite	93 (85.3)	32 (88.9)	15 (78.9)	140 (85.4)	0.51 ^
Amosite and Crocidolite	8 (7.3)	4 (11.1)	3 (15.8)	15 (9.2)	
Amosite	7 (6.4)	0 (0.0)	1 (5.3)	8 (4.9)	
Amosite and Chrysotile	1 (0.9)	0 (0.0)	0 (0.0)	1 (0.6)	
Total	109	36	19	164	

^ Fisher’s exact test.

**Table 10 ijerph-19-14520-t010:** Relationship between asbestos fibre size range and mesothelioma subtype.

Number of Asbestos Fibre Size	Epithelioid *n*, %	Biphasic *n*, %	Sarcomatous *n*, %	Total	*p*-Value ^
Crocidolite 1–5 µm					
0	104 (61.5)	40 (65.6)	23 (71.9)	167 (63.7)	0.64
1–999,999	52 (30.8)	20 (32.8)	9 (28.1)	81 (30.9)	
1,000,000–2,999,999	6 (3.6)	1 (1.6)	0 (0.0)	7 (2.7)	
≥3,000,000	7 (4.1)	0 (0.0)	0 (0.0)	7 (2.7)	
Crocidolite > 5–10 µm					
0	86 (50.9)	35 (57.4)	18 (56.3)	139 (53.1)	0.46
1–999,999	64 (37.9)	24 (39.3)	13 (40.6)	101 (38.6)	
1,000,000–2,999,999	9 (5.3)	2 (3.3)	1 (3.1)	12 (4.6)	
≥3,000,000	10 (5.9)	0 (0.0)	0 (0.0)	10 (3.8)	
Crocidolite > 10 µm					
0	93 (55.0)	33 (54.1)	19 (59.4)	145 (55.3)	0.84
1–999,999	60 (35.5)	25 (41.0)	11 (34.4)	96 (36.6)	
1,000,000–2,999,999	10 (5.9)	3 (4.9)	2 (6.3)	15 (5.7)	
≥3,000,000	6 (3.6)	0 (0.0)	0 (0.0)	6 (2.3)	
Amosite 1–5 µm					
0	162 (95.9)	59 (96.7)	32 (100.0)	253 (96.6)	0.68
1–999,999	6 (3.6)	1 (1.6)	0 (0.0)	7 (2.7)	
1,000,000–2,999,999	1 (0.6)	1 (1.6)	0 (0.0)	2 (0.8)	
≥3,000,000	0 (0.0)	0 (0.0)	0 (0.0)	0 (0.0)	
Amosite > 5–10 µm					
0	159 (94.1)	58 (95.1)	30 (93.8)	247 (94.3)	0.75
1–999,999	9 (5.3)	2 (3.3)	2 (6.3)	13 (5.0)	
1,000,000–2,999,999	1 (0.6)	1 (1.6)	0 (0.0)	2 (0.8)	
≥3,000,000	0 (0.0)	0 (0.0)	0 (0.0)	0 (0.0)	
Amosite > 10 µm					
0	160 (94.7)	57 (93.4)	29 (90.6)	246 (93.9)	0.34
1–999,999	9 (5.3)	3 (4.9)	3 (9.4)	15 (5.7)	
1,000,000–2,999,999	0 (0.0)	1 (1.6)	0 (0.0)	1 (0.4)	
≥3,000,000	0 (0.0)	0 (0.0)	0 (0.0)	0 (0.0)	
Chrysotile 1–5 µm					
0	169 (100.0)	61 (100.0)	32 (100.0)	262 (100.0)	-
1–999,999	-	-	-	-	
1,000,000–2,999,999	-	-	-	-	
≥3,000,000	-	-	-	-	
Chrysotile > 5–10 µm					
0	169 (100.0)	61 (100.0)	32 (100.0)	262 (100.0)	
1–999,999	-	-	-	-	
1,000,000–2,999,999	-	-	-	-	
≥3,000,000	-	-	-	-	
Chrysotile >10 µm					
0	168 (99.4)	61 (100.0)	32 (100.0)	261 (99.6)	1.000
1–999,999	1 (0.6)	0 (0.0)	0 (0.0)	1 (0.4)	
1,000,000–2,999,999					
≥3,000,000					

^ Fisher’s exact test was conducted.

## Data Availability

Data were made available to us by the National Institute of Occupational Health and can be obtained on request and with the relevant approval.

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
