# Peer review of "The Association between the Histological Subtypes of Mesothelioma and Asbestos Exposure Characteristics"

_ijerph, 2022, doi:10.3390/ijerph192114520_

Round 1
Reviewer 1 Report
There are some interesting data in this manuscript but also a lot of data that is excess to requirement and not of great relevance to our understanding of mesothelioma. I would suggest that this study just focus on the asbestos exposure characteristics, such as lung fibre type and number, and employment, and relevant demographics, age and sex, that may influence histological subtypes. This can build on, or add to, the other studies (refs 8 and 31) that have explored this issue. Information on things like province, compensation, and commodities other than asbestos is not necessary because they seem irrelevant to the main issue (factors affecting histological subtypes). The number of results, including tables, should be reduced to just the relevant variables.
It is interesting why greater than a third of lung biopsies, including a third of occupational cases, had no asbestos fibres identified. No reason for this is offered. Were the cases without fibres identified more likely to be exposed to chrysotile, although chrysotile is also likely to be biopersistant (see Feder et al. The asbestos fibre burden in human lungs: new insights into the chrysotile debate. ERJ 2017; 49: 1602534)?
Minor points:
1. 95%CI are not necessary when presenting the proportion of cases environmentally and occupationally exposed (lines 127 – 128). These are actual numbers, not estimates.
2. Line 140, remove ‘As might be expected..’ as this is a results section, which doesn’t require comment
3. Lines 150 – 152 are repetition of lines 127/128
Author Response
Response to Reviewer 1 Comments
Thank you for taking the time to review our article. We hope we have addressed all your comments and thank you for helping us to improve the article.
Point 1
There are some interesting data in this manuscript but also a lot of data that is excess to requirement and not of great relevance to our understanding of mesothelioma. I would suggest that this study just focus on the asbestos exposure characteristics, such as lung fibre type and number, and employment, and relevant demographics, age and sex, that may influence histological subtypes. This can build on, or add to, the other studies (refs 8 and 31) that have explored this issue. Information on things like province, compensation, and commodities other than asbestos is not necessary because they seem irrelevant to the main issue (factors affecting histological subtypes). The number of results, including tables, should be reduced to just the relevant variables.
Response 1
Although the compensation outcomes contribute to the story for the South African reader, I do agree that it might shadow the main aim and is confusing to international readers. The compensation portion was removed.
Point 2
It is interesting why greater than a third of lung biopsies, including a third of occupational cases, had no asbestos fibres identified. No reason for this is offered. Were the cases without fibres identified more likely to be exposed to chrysotile, although chrysotile is also likely to be biopersistant (see Feder et al. The asbestos fibre burden in human lungs: new insights into the chrysotile debate. ERJ 2017; 49: 1602534)?
Response 2
Added a paragraph in the discussion starting from line 328. “No asbestos fibres were identified for 37.4% of cases. Some of these lungs were described as completely being replaced by tumour where in others only asbestos bodies were identified.”
Point 3
Minor points:
- 95%CI are not necessary when presenting the proportion of cases environmentally and occupationally exposed (lines 127 – 128). These are actual numbers, not estimates.
- Line 140, remove ‘As might be expected.’ as this is a results section, which doesn’t require comment
- Lines 150 – 152 are repetition of lines 127/128
Response 3
- This was removed.
- This was removed.
- This was removed in line 150-152.
Reviewer 2 Report
This most interesting and original paper deserves publication. However, it requires some revisions (and shortenings).
To the International audience of the journal, informations regarding compensation are difficult to understand. How relevant are they to the purpose of this article? Are they really necessary in this article? The authors omit any mention to this theme in the conclusions.
Major comments
· The representativeness of mesotheliomas investigated in this study over all mesotheliomas occurring in South Africa during 2006-2016 is not clear.
· In more than one third of the lungs of occupationally exposed cases of mesothelioma, no fibres were identified (Table 5). This is most interesting This proportion is higher than that found in the general British population ( Gilham et al, Occup Environ Med 2016;73:290-299). How appropriate were the methods which have been used? If methods were appropriate, a plausible interpretation of the high poportion of “negative” finding is that all or most of such cases had been exposed to chrysotile, which is much less persistent than the amphiboles. Authors should express an opinion on this point.
· Further, authors should add some information of the distribution of lung findings by type of asbestos exposure suggested by occupational anamneses.
· Paragraph 3.4 “Compensation outcomes” and tables 3 and 4 are difficult to understand by non South Africans. Same applies to corresponding paragraphs in the discussion (lines 300-312). If authors intend to describe unevenesses in access to recognitions, they should provide many more details (perhaps, in a separate publication).
· Is it really necessary to present p-values with 3 or 4 decimals?
Minor observations..
Introduction
· Line 43: why censoring latency at 40 years? Mesotheliomas may be diagnosed even later
· Lnes 62-63: some additional details about ART and KRT would be interesting: how are these trusts funded? Which proportion of mesotheliomas in SA have access to these trusts?
Study population and data collection
· Lines 74-76: Be more precise about the cases included in the database of present paper. All mesothelioma cases diagnosed in SA between 2006 and 2016? All deceased miners, ex miners and ART/KRT claimants? Cases submitted to autopsy ? Diagnosed cases? What is the proportion of cases non included in the PATHAUT? Readers of the paper should be informed on size of any selection of cases and possible ensuing bias.
· Lines 81-84: some more details about (or at least a reference to) methods used in the EM Unit for counting and measuring fibres in the lung samples are needed. Same comment about asbestos body counts.
Variable descriptions and data analysis
· Lines 100-102: provide criteria for establishing cutting points at 5 and 10 milimicrons. Similar question for fibre concentrations (why didn’t authors use tertiles?)
· Lines 141-142: Describe circumstances and types of “environmental exposure”. Authors are right in expecting longer exposures in the environmental cases. Nevertheless, among environmental cases, average duration of exposure (23.5 years) is much shorter than average age at diagnosis (lines 131-132).
Results: demographic data
· Line 127: did the 270 cases represent all cases of mesotelioma in SA newly diagnosed in 2006-2016?
· Line 129: for no case there was no anamnestical evidence of exposure to asbestos: was it so? This is interesting in relation to the debate on whether or not mesothelioms can arise independently of any exposure to asbestos. This is why information on any selection of cases is important.
Results: exposure data
- Lines 155-156; I do not understand what authors mean stating that “The mean length ….. double digits” in Table 2.
- Line 159: in the legend of Table 2, specify that “exposure” is “Exposure to asbestos”
Results: compensation outcomes
- The non-SA reader has some problems in understanding this section. Why as many as 45.6% are unknown to NIOH? Given the high proportion of “unknown”, perhaps this section could be eliminated.
Results: fibre burden
- Lines 179-184: text is repetitious of the contents of Table 5. Is it really necessary?
Results: histological features
- Lines 186-209:Again, I recommend the authors to avoid unnecessary repetitions between text and tables 6 and 7
Results: fibre burden per subtype
- Line 217: in Table 8, why is the mean concentration of total asbestos fibres much higher then the sum of means of specific asbestos types? In addition, the table would provide more information if absolute numbers of cases were reported in each cell,
- Line 218: same comment for Table 9.
Discussion and conclusions
· Lines 270-279: authors should be aware of the unappropriateness of inferences based on the comparison of age between case-only series (Consonni D, Barone-Adesi F, Mensi C. Br J Cancer. 2014 Oct 14;111(8):1674)
· Line 317: [2525]???
· Lines 317-321: authors should provide their interpretation of the fact that one third of the lung speciments did not contain fibres. What is the evidence that these cases were exposed to chrysotile?
· Lines 345-347: what do authors mean with the term “evolution”? In terms of prognosis, the present paper does not provide any new insight. In terms of risk, their statement is probably correct, but they should comment on the fact than no fibres were found in one third of their lung specimens: is this compatible with the well known persistence of the amphiboles? If these cases cannot be attributed to amphiboles, how should their etiology by interpreted?
· Line 357: allusion to prognosis is out of place. Present study did not investigate survival.
· Line 358: Does the sharp statement that “the aetiology of mesotelioma subtyipes remains unknown” imply that present study does not provide any new information on the association between asbestos type, fibre size and burden and mesotelioma subtype (see introduction, lines 69-71)?
Author Response
Response to Reviewer 2 Comments
Thank you for taking the time to review our article. We hope we have addressed all your comments and thank you for helping us to improve the article.
Point 1
This most interesting and original paper deserves publication. However, it requires some revisions (and shortenings).
To the International audience of the journal, informations regarding compensation are difficult to understand. How relevant are they to the purpose of this article? Are they really necessary in this article? The authors omit any mention to this theme in the conclusions.
Response 1
Although the compensation outcomes contribute to the story for the South African reader, I do agree that it might shadow the aim and is confusing to international readers. The compensation portion was removed.
Major comments
Point 2
The representativeness of mesotheliomas investigated in this study over all mesotheliomas occurring in South Africa during 2006-2016 is not clear.
Response 2
Added this paragraph in the discussion section, starting in line 364. “The mesothelioma cases extracted from the PATHAUT database over the 11-year period only presents cases that came through the NIOH for diagnosis as part of the compensation process. A study reported trends of mesothelioma cases among South Africans during a similar period and cases reported in PATHAUT equates about 18% of the overall mesothelioma cases in South Africa. There are several barriers to requesting and providing consent for autopsy by relatives of deceased miners. This was previous described in 2017.”
Point 3
In more than one third of the lungs of occupationally exposed cases of mesothelioma, no fibres were identified (Table 5). This is most interesting. This proportion is higher than that found in the general British population (Gilham et al, Occup Environ Med 2016;73:290-299). How appropriate were the methods which have been used? If methods were appropriate, a plausible interpretation of the high poportion of “negative” finding is that all or most of such cases had been exposed to chrysotile, which is much less persistent than the amphiboles. Authors should express an opinion on this point.
Response 3
Added this paragraph in the discussion section starting in line 327. “No asbestos fibres were identified for 37.4% of cases. Some of these lungs were described as completely being replaced by tumour where in others only asbestos bodies were identified.”
Point 4
Further, authors should add some information of the distribution of lung findings by type of asbestos exposure suggested by occupational anamneses.
Response 4
All cases are of deceased miners and their exposure history are from records of previous mines obtained from the deceased’s relative/s that forms part of the compensation process.
Point 5
Paragraph 3.4 “Compensation outcomes” and tables 3 and 4 are difficult to understand by non-South Africans. Same applies to corresponding paragraphs in the discussion (lines 300-312). If authors intend to describe unevenesses in access to recognitions, they should provide many more details (perhaps, in a separate publication).
Response 5
This has been covered in Response 1, copied here: Although the compensation outcomes contribute to the story for the South African reader, I do agree that it might shadow the aim and is confusing to international readers. The compensation portion was removed.
Point 6
Is it really necessary to present p-values with 3 or 4 decimals?
Response 6
Rounded up to 2 decimals.
Point 7
Minor observations.
Introduction
- Line 43: why censoring latency at 40 years? Mesotheliomas may be diagnosed even later
This is only given as an indication and it is know mesotheliomas can occur either much earlier or much later.
- Lines 62-63: some additional details about ART and KRT would be interesting: how are these trusts funded? Which proportion of mesotheliomas in SA have access to these trusts?
The ART was formed in 2003, through a class action settlement against the Cape Public Limited Company (Plc) and the General Mining Corporation (Gencor). This settlement made provision for claimants and environmental. The KRT was formed in 2006, through a legal settlement against the Swiss Eternit Group. The Trust has ena-bled ex-miners of the Kuruman and Danielskuil Cape Blue Asbestos (KCBA and DCBA) mines to apply for compensation. The story of the ART/KRT was described in 2014 by J teWaterNaude.
Study population and data collection
- Lines 74-76: Be more precise about the cases included in the database of present paper. All mesothelioma cases diagnosed in SA between 2006 and 2016? All deceased miners, ex miners and ART/KRT claimants? Cases submitted to autopsy ? Diagnosed cases? What is the proportion of cases non included in the PATHAUT? Readers of the paper should be informed on size of any selection of cases and possible ensuing bias.
This has been covered in Response 2, copied here: Added this paragraph in the discussion section, starting in line 364. “The mesothelioma cases extracted from the PATHAUT database over the 11-year period only presents cases that came through the NIOH for diagnosis as part of the compensation process. A study reported trends of mesothelioma cases among South Africans during a similar period and cases reported in PATHAUT equates about 18% of the overall mesothelioma cases in South Africa. There are several barriers to requesting and providing consent for autopsy by relatives of deceased miners. This was previous described in 2017.”
- Lines 81-84: some more details about (or at least a reference to) methods used in the EM Unit for counting and measuring fibres in the lung samples are needed. Same comment about asbestos body counts.
This paragraph was added to the methods section: "At the NIOH, the lung fibre burden is determined by extracting the asbestos fibres from the lungs. The asbestos fibre types, fibre sizes and asbestos body concentration are determined by SEM together with Energy Dispersive Spectroscopy (EDS). This method was previously described by Phillips and Murray 2010 [16]."
Variable descriptions and data analysis
- Lines 100-102: provide criteria for establishing cutting points at 5 and 10 milimicrons. Similar question for fibre concentrations (why didn’t authors use tertiles?)
The laboratory established the fibre sizes as 1-5, 5-10 and 10>. > 1 million asbestos fibres per dried lung content is regarded as significant exposure.
- Lines 141-142: Describe circumstances and types of “environmental exposure”. Authors are right in expecting longer exposures in the environmental cases. Nevertheless, among environmental cases, average duration of exposure (23.5 years) is much shorter than average age at diagnosis (lines 131-132).
Data concerning initial diagnosis is not recorded in the PATHAUT database. Added paragraph (paragraph 5) in the introduction explaining environmental exposure: "Exposure to asbestos can be grouped into occupational and non-occupational exposure. Occupational exposure includes asbestos miners, millers and workers involved in the manufacturing of asbestos products. Non-occupational exposure may be grouped into domestic, neighbourhood and true environmental exposure. Domestic exposure can also be referred to as para-occupational or familial exposure. This occurs when asbestos workers carry asbestos fibres home on their working clothes which usually affects their family members. Neighbourhood exposure, also referred to as environmental exposure, affects residents living close to mine tailings or other asbestos contaminated areas. True environmental exposure arises from naturally occurring asbestos contaminated soil [13]."
Results: demographic data
- Line 127: did the 270 cases represent all cases of mesotelioma in SA newly diagnosed in 2006-2016?
No, diagnosed when deceased during autopsy as part of the compensation process only.
- Line 129: for no case there was no anamnestical evidence of exposure to asbestos: was it so? This is interesting in relation to the debate on whether or not mesothelioms can arise independently of any exposure to asbestos. This is why information on any selection of cases is important.
All cases reported were deceased miners or ex-miners. The exposure data is based on the longest service history.
Results: exposure data
- Lines 155-156; I do not understand what authors mean stating that “The mean length ….. double digits” in Table 2.
The number of occupational cases with the longest service history in asbestos is >100 whereas other commodities are less than 100.
- Line 159: in the legend of Table 2, specify that “exposure” is “Exposure to asbestos”
This has been covered in point 8 above, copied here: All cases reported were deceased miners or ex-miners. The exposure data is based on the longest service history.
Results: compensation outcomes
- The non-SA reader has some problems in understanding this section. Why as many as 45.6% are unknown to NIOH? Given the high proportion of “unknown”, perhaps this section could be eliminated.
This has been covered in Response 1, copied here: Although the compensation outcomes contribute to the story for the South African reader, I do agree that it might shadow the aim and is confusing to international readers. The compensation portion was removed.
Results: fibre burden
- Lines 179-184: text is repetitious of the contents of Table 5. Is it really necessary?
The outcome of response 1 led this to being removed.
Results: histological features
- Lines 186-209: Again, I recommend the authors to avoid unnecessary repetitions between text and tables 6 and 7
In Table 6, histological features by exposure type are described. In Table 7, it is further broken up in age and sex. No repetitions.
Results: fibre burden per subtype
- Line 217: in Table 8, why is the mean concentration of total asbestos fibres much higher then the sum of means of specific asbestos types? In addition, the table would provide more information if absolute numbers of cases were reported in each cell.
- Line 218: same comment for Table 9.
We will not have enough time to add this. It is noted for future publications.
Discussion and conclusions
- Lines 270-279: authors should be aware of the unappropriateness of inferences based on the comparison of age between case-only series (Consonni D, Barone-Adesi F, Mensi C. Br J Cancer. 2014 Oct 14;111(8):1674)
Thank you for the interesting information. Seems like this study makes a specific reference to latency. We reported on age at death for mesothelioma cases and not latency.
- Line 317: [2525]???
Corrected
- Lines 317-321: authors should provide their interpretation of the fact that one third of the lung speciments did not contain fibres. What is the evidence that these cases were exposed to chrysotile?
This has been covered in Response 3, copied here: Added this paragraph in the discussion section starting in line 327. “No asbestos fibres were identified for 37.4% of cases. Some of these lungs were described as completely being replaced by tumour where in others only asbestos bodies were identified.”
- Lines 345-347: what do authors mean with the term “evolution”? In terms of prognosis, the present paper does not provide any new insight. In terms of risk, their statement is probably correct, but they should comment on the fact than no fibres were found in one third of their lung specimens: is this compatible with the well known persistence of the amphiboles? If these cases cannot be attributed to amphiboles, how should their etiology by interpreted?
Evolution was removed. This has been covered in Response 3, copied here: Added this paragraph in the discussion section starting in line 327. “No asbestos fibres were identified for 37.4% of cases. Some of these lungs were described as completely being replaced by tumour where in others only asbestos bodies were identified.”
- Line 357: allusion to prognosis is out of place. Present study did not investigate survival. Removed.
- Line 358: Does the sharp statement that “the aetiology of mesotelioma subtyipes remains unknown” imply that present study does not provide any new information on the association between asbestos type, fibre size and burden and mesotelioma subtype (see introduction, lines 69-71)?
When read together with this sentence it places it in perspective. “No strong evidence was found to support any relationship between mesothelioma subtype and asbestos type, fibre size or asbestos burden.”
Reviewer 3 Report
Thanks for giving me to review this manuscript. Authors described data collected on individuals diagnosed with mesothelioma by autopsy in South Africa. In addition, they explored the associations between the asbestos type, fibre size and burden and mesothelioma subtypes. This is an interesting manuscript. However, there are several methodological aspects the warrant further clarification and alternative discussion may be indicated. There are so many outcomes such as asbestos exposure, histological subtypes, and compensation, and so many exposures including demographic data and asbestos measurement. If authors want to say " No strong evidence was found to support any relationship between mesothelioma subtype and asbestos type, fibre size or asbestos burden". They should limit the outcome and organize the flow of discussion to conclusion. How did authors (NIOH) evaluate the source of occupational and environmental exposures? Authors should explain in details to clarify the bias that would be inherent in the measurement. Authors explained some in the 2nd paragraph of Discussion. However, the operational definition is unclear. 3.1. Demographic data Authors should avoid using confidence intervals in the description. It is because the population were not sampled in random manner. Table 2 What does the *Other mean? From the results, authors can say most tissue of MPM contained crocidolite. However, they can not conclude "crocidolite is the main cause of mesothelioma". It is because authors did not evaluate those who did not suffer MPM. Population attributable risk was not clear.Author Response
Response to Reviewer 3 Comments
Thank you for taking the time to review our article. We hope we have addressed all your comments and thank you for helping us to improve the article.
Comments and Suggestions for Authors
Thanks for giving me to review this manuscript. Authors described data collected on individuals diagnosed with mesothelioma by autopsy in South Africa. In addition, they explored the associations between the asbestos type, fibre size and burden and mesothelioma subtypes. This is an interesting manuscript. However, there are several methodological aspects the warrant further clarification and alternative discussion may be indicated. There are so many outcomes such as asbestos exposure, histological subtypes, and compensation, and so many exposures including demographic data and asbestos measurement. If authors want to say, " No strong evidence was found to support any relationship between mesothelioma subtype and asbestos type, fibre size or asbestos burden". They should limit the outcome and organize the flow of discussion to conclusion.
Point 1
How did authors (NIOH) evaluate the source of occupational and environmental exposures? Authors should explain in details to clarify the bias that would be inherent in the measurement.
Response 1
This was evaluated upon obtaining information for the employment history. Cases that applied to the ART/KRT for environmental exposure compensation would be received by the NIOH in such a manner.
Point 2
Authors explained some in the 2nd paragraph of Discussion. However, the operational definition is unclear. 3.1. Demographic data Authors should avoid using confidence intervals in the description. It is because the population were not sampled in random manner.
Response 2
Confidence intervals were removed.
Point 3
Table 2 What does the *Other mean? From the results, authors can say most tissue of MPM contained crocidolite. However, they cannot conclude "crocidolite is the main cause of mesothelioma". It is because authors did not evaluate those who did not suffer MPM. Population attributable risk was not clear.
Response 3
*Other description added. Agreed, paragraph removed.
Round 2
Reviewer 1 Report
The authors have adequately addressed my previous comments. The only minor change I would suggest is moving the opening paragraph of the methods and combine it with the details of asbestos lung fibre burden (second paragraph of section 2.1
Author Response
Reviewer 1 comments – Round 2
The authors have adequately addressed my previous comments. The only minor change I would suggest is moving the opening paragraph of the methods and combine it with the details of asbestos lung fibre burden (second paragraph of section 2.1)
Response
The opening paragraph was incorporated into section 2.1.

Reviewer 2 Report
What information was available (and used) to classify cases as occupationally or environmentally exposed to asbestos?
Line 251: Cases included in this paper represent 18% of all cases in South Africa and are not necessarily of all cases in tje ,population. This ought to be mentioned in the abstract
In one third of the cases no asbestos fibres were found in the lung parenchyma. According to the authors’ reply to my previous comments, in some of these cases, no lung parenchyma was spared by the tumour. It would still be desirable to know whether there were cases (and how many) in which the evidence for lack of any asbestos fibers was convincing. If there are, is there any evidence that such cases may have been exposed exclusively to chrysotile?
The final but one statement (lines 318-322) “The aetiology of these subtypes remains unknown” does not reflect the message conveyed by this paper and should be reworded. This study, as many others, shows that the aetiology of all histological subtypes of mesotelioma is largely known and that asbestos may cause all of them. Additional research envisaged by Authors should be specified more precisely.
Author Response
Reviewer 2 comments – Round 2
- What information was available (and used) to classify cases as occupationally or environmentally exposed to asbestos?
- Line 251: Cases included in this paper represent 18% of all cases in South Africa and are not necessarily of all cases in tje ,population. This ought to be mentioned in the abstract
- In one third of the cases no asbestos fibres were found in the lung parenchyma. According to the authors’ reply to my previous comments, in some of these cases, no lung parenchyma was spared by the tumour. It would still be desirable to know whether there were cases (and how many) in which the evidence for lack of any asbestos fibers was convincing. If there are, is there any evidence that such cases may have been exposed exclusively to chrysotile?
- The final but one statement (lines 318-322) “The aetiology of these subtypes remains unknown” does not reflect the message conveyed by this paper and should be reworded. This study, as many others, shows that the aetiology of all histological subtypes of mesotelioma is largely known and that asbestos may cause all of them. Additional research envisaged by Authors should be specified more precisely.
Response
- The family of the deceased consents to the ART/KRT for the collection of information on the deceased. The consent asks for details of the deceased like ID number and a full work history. Once submitted, an enquiry is lodged to confirm the work history of the deceased. If the deceased meets the criteria for compensation through the ART/KRT a post-mortem will be completed by the National Institute for Occupational Health (NIOH). The family of the deceased consents to the post-mortem prosection using the consent forms of the NIOH. This requires the family to indicate that the deceased was a miner/ex-miner. An enquiry is lodged to obtain/confirm the work history of the deceased. Reports are submitted to the Medical Bureau for Occupational Diseases (MBOD) for compensation under the Occupational Diseases in Mines and Works Act 78 of 1973 (ODMWA, 1973). If the deceased meets the criteria for compensation the case will be submitted to the ART/KRT. If the deceased was a miner/ex-miner, the case will be classified as occupational.
Environmental cases are only submitted through the ART/KRT. The consent requires information such as if the deceased lived with anyone who worked at a qualifying operation, life events and residential history as well as a full employment history. If the deceased meets the qualifying criteria, the case will be classified as an environmental case.
The ART/KRT filed any existing work records from the founding companies as they recognised that claimants would lack resources. The ART/KRT criteria was described in detail in TeWaterNaude, J. M. "The story of the Asbestos Relief Trust-Part 2." Occupational Health Southern Africa (2014).
- The following sentence was added to the abstract “The study population does not include all cases of mesothelioma in South Africa, but rather those that reached the compensation system”.
- While the topic “does chrysotile alone cause mesothelioma” is very important, the authors believe this question raises a separate study, as the authors cannot say whether any of these subjects were exposed exclusively to chrysotile as we do not have this data. Our only possible explanation is that the subject may have cleared asbestos from the lungs during life.
- The following sentence was removed from the conclusion: “The aetiology of these subtypes remain unknown.”

Reviewer 3 Report
Authors responded all of my comments.
Author Response
Reviewer 3 comments – Round 2
Authors responded all of my comments.
No response needed.